# Positron Annihilation Study of RPV Steels Radiation Loaded by Hydrogen Ion Implantation

**DOI:** 10.3390/ma15207091

**Published:** 2022-10-12

**Authors:** Vladimir Slugen, Tomas Brodziansky, Jana Simeg Veternikova, Stanislav Sojak, Martin Petriska, Robert Hinca, Gabriel Farkas

**Affiliations:** 1Institute of Nuclear and Physical Engineering, Faculty of Electrical Engineering and Information Technology, Slovak University of Technology, Ilkovicova 3, 812 19 Bratislava, Slovakia; 2Advanced Technologies Research Institute, Faculty of Materials Science and Technology, Slovak University of Technology, Jana Bottu 25, 917 24 Trnava, Slovakia

**Keywords:** VVER reactor, long-term operation, radiation-induced defects, recovery annealing, positron annihilation techniques

## Abstract

Specimens of 15Kh2MFAA steel used for reactor pressure vessels V-213 (VVER-440 reactor) were studied by positron annihilation techniques in terms of their radiation resistance and structural recovery after thermal treatment. The radiation load was simulated by experimental implantation of 500 keV H^+^ ions. The maximum radiation damage of 1 DPA was obtained across a region of 3 µm. Radiation-induced defects were investigated by coincidence Doppler broadening spectroscopy and positron lifetime spectroscopy using a conventional positron source as well as a slow positron beam. All techniques registered an accumulation of small open-volume defects (mostly mono- and di-vacancies) due to the irradiation, with an increase of the defect volume ΔV_D_ ≈ 2.88 × 10^−8^ cm^−3^. Finally, the irradiated specimens were gradually annealed at temperatures from 200 to 550 °C and analyzed in detail. The best defect recovery was found at a temperature between 450 and 475 °C, but the final defect concentration of about ΔC_D_ = 0.34 ppm was still higher than in the as-received specimens.

## 1. Introduction

Prolongation of nuclear power plants’ (NPPs) lifetime, also known as long-term operation (LTO), allows NPPs to generate reliable, low-cost, low-emission electricity for many years longer than originally envisioned and thus maximize their value. Today’s operating nuclear reactors were originally designed for 30 to 40 years of operation, but there is no fixed technical limit to the life of the reactors [1]. LTO of NPPs has been successfully demonstrated and is increasingly recognized internationally as standard practice. LTO of an NPP may be conditioned by life-limiting processes and features of structures, systems, and components—the emphasis is on the irreplaceable reactor pressure vessel (RPV) and its construction steel. Degradation of its mechanical properties is the crucial limiting factor of NPP lifespan [2]. Elements causing degradation of RPV properties are neutron irradiation, high temperature and pressure, fatigue, corrosion, etc. Therefore, it is recommended to evaluate periodically the structural integrity of an RPV and to predict the future development of RPV degradation. It is well-known that neutron irradiation is the main driver of the microstructural changes in the RPV during reactor operation. RPV steels that are exposed to a wide spectrum of irradiation (including fast neutrons, whose influence is dominant) during LTO will degrade over time via effects connected to radiation damage. The traditional method of studying microstructural changes in reactor steels is based on stay (irradiation) in material research reactors, where (depending on position) irradiation of special model specimens is accelerated. Of course, although accelerated, these experiments take several months or years and cannot be equal in complexity to real conditions in commercial NPPs. Additionally, it is necessary to note that these experiments are not cheap and are difficult from a handling and radiation protection point of view. A promising approach could be the application of light ion accelerators for experimental simulation of neutron irradiation [3] performed on original commercial RPV steels.

In this paper, we will be focused mainly on positron annihilation techniques, which have been frequently applied in nuclear material microstructural studies in the last 50 years. Detailed descriptions of these techniques have been published worldwide in numerous scientific publications [4,5,6,7]. In the last 30 years, our laboratory has focused on the application of positron annihilation spectroscopy (PAS) techniques to nuclear reactor pressure vessel steels. In particular, RPV steels of Russian water-moderated and water-cooled pressured reactors (VVERs) were at the centrum of our studies. Although during the years we have come to understand the limits of these techniques, the conclusions and results interpretations we have made have resulted from combination with results from other techniques, and we continue with our trials to upgrade the level of experimental resolution as well as minimize the influence of possible disturbing factors as much as possible. An application of proton implantation as an experimental simulation of neutron irradiation seems appropriate in this study.

Nevertheless, we have touched also on point of limited reproducibility of results due to material inhomogeneities and a wide range of different possible errors.

### Degradation of Reactor Pressure Vessel Steels

At the end of 2020, worldwide, 292 reactors have been in operation for longer than 30 years. This represents more than 65% of power reactors. During the last decade, only 63 new units were put in operation. From a long-term perspective and regarding the projected design lifetime, in 2030 about 160 reactors will be shut down [8,9]. Refurbishments and LTO seem to be a clever way to extend NPPs’ lifetime and minimize global warming’s impact on the world. It is important to stress that the operation of NPPs can deliver baseload power and can give a guarantee of quite acceptable and stable electricity costs.

Several NPPs, most notably 73 units in the United States, prolonged their operational license for up to 60 years. A similar trend is also present in other countries, where the crucial condition for operational lifetime prolongation is a decision from periodic safety reviews (PSRs). Based on these PSRs, which are supervised by a national nuclear regulatory authority, factors such as (i) aging management, (ii) environmental management, (iii) operating experiences, (iv) safety and security improvements, and emerging issues are considered and analyzed in detail [10].

The essential mechanisms of RPV irradiation embrittlement were deeply studied and analyzed via different RPV surveillance specimen programs. The RPV steel specimens were placed in specially designed capsules and irradiated in an operating reactor at exact analyzed conditions for several years. The base for evaluations was tensile, Charpy-V, and fracture toughness tests. After testing, these specimens were cut and used for additional non-destructive testing via different spectroscopic methods, and thanks to placement closer to the reactor core, an accelerating factor of up to 10 could be achieved [11].

Generally, it was stated that the most important embrittlement mechanisms are direct matrix damage, formation, and movement of Cu-rich precipitates (reported as CRP, with connection to Ni, Mn, Si, etc.), MnNi-rich precipitates (MNP), and P segregation on grain boundaries [12,13]. Recently published papers [14,15] also described Cu- and Ni/Si/Mn-rich clusters in irradiated reactor steels and they are often being located near line dislocations. Structural defects such as vacancies or dislocations probably can suppress the formation of large Ni/Si/Mn-rich and Cu-rich clusters by reducing the diffusion process under irradiation.

The neutron flux during the operation of commercial reactors depends on their design as well as power. Generally, in pressurized water reactors (PWRs), the neutron flux ranges between 10^9^ and 10^12^ n.cm^−2^.s^−1^. The real impact of neutrons on irradiation damage in this relatively wide range of fluxes is influenced also by its spectrum, where special importance is placed on the fast neutrons (basically E > 1 MeV, in the case of VVERs neutrons with E > 0.5 MeV being considered) and differences in total neutron fluences. Of course, differences in coolant temperature (in the case of PWRs 260 to 330 °C), water chemistry, and reactor operation dynamics (out of all thermal pressurized shocks) can play a special role in real design material embrittlement. With an increase in radiation exposure, an increase in the number of obstacles can be observed too, and higher stresses are required for dislocation motion, with a resulting increase in the yield strength of the material. We have to consider also non-hardening embrittlement, which can appear via radiation-induced solute segregation to grain boundaries. This effect is usually assigned to the segregation of phosphorous on the grain boundary [16]. Many studies performed in the last 50 years have confirmed that the dominant effect on embrittlement from alloying elements involves copper. Its content dramatically influences irradiation sensitivity, more so than nickel, phosphorus, and vanadium, which are considered important contributors as well [11,16]. In addition to these elements, sulfur content, due to its negative influence on RPV steels’ resilience, should be strictly controlled.

The duration of operational RPV lifetime is limited by neutron embrittlement. The crucial effect is resistance against potential brittle damage. Prolongation of NPP operation from the previously scheduled 40 years to the considered 60–80 years implies an increasing neutron exposure of about 100%. This lifetime extension could increase the maximal level of neutron fluences up to 1 × 10^20^ n.cm^−2^ [17]. The neutron irradiation shifts the Charpy toughness curve to increased levels of temperature. The second characteristic of the effect of increased irradiation is a significant decrease in fracture resistance [18].

One effective way to recover (at least partially) fracture toughness properties decreased due to irradiation of RPV steels is thermal annealing. Although we use the term annealing, it is correct to note that temperatures lower than 500 °C are not enough for complex annealing. The restoration of material toughness via thermal annealing was the focus of nuclear utilities mostly in connection to the VVER-440 V-230 design due to the not-so-strictly limited content of Cu and P in 15Kh2MFA steels, and annealing treatment was an acceptable solution for the increase of the operational safety margin as well as the pressurized thermal shock criteria requested by national regulatory authorities [19,20].

It is necessary to note that there exists a substantial difference between the old generation of the VVER-440 type V-230 and the second generation, called the V-213, regarding the chemical composition of steels. Therefore, the second generation, with minimized content of Cu and P, is marked as 15Kh2MFAA [21]. The differences are shown in Table 1, and an illustrative cross-section of typical VVER-440 RPV steel is depicted in Figure 1.

Based on results from several materials, studies of RPV steel specimens focused on macroscopical as well as microscopical changes due to annealing; the crucial conclusion was made from the tensile notch and Charpy-V tests. In the case of wet annealing, the knowledge was summarized in [22,23]. Changes in microstructure were observed mostly via transmission electron microscopy with relatively low resolution (corresponding to the time about 1980) at temperatures of ~340 °C minimum. This low temperature is an essential obstacle preventing significant minimization of irradiation embrittlement. Based on [23], the recovering effect is less than 50%. On the other hand, for so-called dry annealing with a much higher difference (≈230 °C) from the operating temperature at which RPV steels were long-term irradiated, the effectiveness level is over 90% [24,25]. The recent electron microscopy investigation of RPV steels and Fe-Cr model alloys after irradiation and annealing are described in more detail in [26,27,28].

Especially important approaches for evaluation of RPV steels’ long-term operational degradation were surveillance specimen programs [29,30], which provided the possibility to observe continuously the shift of ductile-brittle transmission temperature (DBTT) curves. Additionally, broken Charpy-V specimens were suitable for the preparation of new specimens for non-destructive testing via different methods of studying the phenomena but could contribute to the complex information about neutron embrittlement as well as the effectiveness of thermal annealing of RPV base and weld materials. In the last decade of the previous century, several spectroscopic methods (including positron annihilation techniques, whose applications will be reported in the next chapters) were used in these studies with the aim of describing microstructural changes, defect creation/relaxation in annealing, the role of alloying elements or long-term thermal and radiation treatment. From the positron annihilation sensitivity on vacancy-type defects point of view, irradiation damage studies were dominant.

From the complexity of irradiation that impacts RPV steels during decades of operation, we select neutron irradiation. Of course, there are also beta and gamma particles of different spectra or energies, but in comparison to the high-energy neutrons, we consider it plausible to neglect them. Similarly, alpha particles or potential light ions play no role due to distance from the core. Therefore, neutron irradiation is dominant in all considerations. The limiting factor of most conventional studies and methods is the activation of the specimens. Handling (cutting, surface cleaning) and possible contamination require special devices including hot cells, which make these studies expensive, time-consuming, and difficult from the radiation protection point of view. Experiments in material test reactors could not always achieve the requested conditions, mostly in neutron fluxes, fluences, temperatures, or irradiation dynamics.

Computer simulations were and still are frequently used in the last few decades but cannot replace material studies in an adequate form. Results from these simulations are sometimes unique, but at least some of them should be experimentally verified. There are fewer research reactors and, due to the limited possibility for experiments on power reactors, also less chance for verification. Therefore, experimental simulation via ion implantation (ion irradiation) seems to be one of the proper methods to conduct this type of study in the future. We have already started bearing in mind that verifying the equivalency of neutron and ion irradiation is a very important task [31,32,33]. For experimental neutron treatment, we have used proton (H^+^) implantations due to our 0.5 MeV cascade accelerator.

There is no doubt that the degree of RPV embrittlement in a reactor pressure vessel (RPV) is a complex function of different parameters such as temperature, neutron fluence, flux, material chemistry, etc. [34,35,36,37,38,39,40].

## 2. Methodology

From the available positron annihilation techniques, we applied positron annihilation lifetime spectroscopy (PALS), coincidence Doppler broadening spectroscopy (CDBS), and a pulsed low-energy positron system (PLEPS).

Generally, the lifetimes of positrons, which are trapped in monovacancies, determined both experimentally and theoretically lie in the interval of 150–300 ps for metals [41]. The lifetimes of positrons trapped in small vacancy clusters have been calculated and experimentally verified many times. As the lattice relaxation around the cluster has a relatively small influence on positron annihilation parameters, it was not included in the calculations. The results show that the lifetime of a positron trapped by di-vacancy does not differ much from that for monovacancy (an increase is about 10 ps for Fe). The lifetime increases rapidly when the cluster grows into two-dimensional tri-vacancy and further into three-dimensional tetra-vacancy [41]. For very large clusters (over 50 vacancies), the lifetime saturates around 450–500 ps. For large voids, the trapping may be limited mainly by positron diffusion to the defect.

Theoretical calculations have revealed that the dislocation line is only a shallow trap for positrons [42,43] (binding energy <0.1 eV). On the other hand, the lifetimes of trapped positrons observed in plastically deformed metals [44] are only slightly lower than the lifetimes of positrons trapped in vacancies. Smedskjaer [45] suggested that the pure dislocation line is a weak positron trap and explained the long lifetimes seen in experiments by point-like defects (vacancies, jogs) associated with the dislocation. For example, the binding energy of 0.92 eV for a vacancy to the edge-dislocation line was calculated in [46] for Fe. Once a positron arrives at the core of the dislocation, it diffuses very quickly (pipe diffusion) until it finds a vacancy attached to the dislocation or a jog of the dislocation; it is then trapped and annihilates there. This explanation is supported also by further calculations [43,47,48,49]. Calculated lifetimes of positrons trapped in dislocations and corresponding binding energies are listed in Table 2.

The specific trapping rates ν_D_ for dislocations were obtained by a combination of positron lifetime measurement and transmission electron microscopy (TEM) or other techniques capable of determining dislocation density (e.g., X-ray diffraction profile) [49]. The values of ν_D_ lie in the range of about 10^−5^–10^−4^ m^2^s^−1^ for metals. Thermally activated de-trapping of positrons trapped in a dislocation core (initial shallow trap) may occur at elevated temperatures, which makes ν_D_ temperature-dependent [50].

Table 2 shows calculated lifetimes τ (ps) of positrons trapped in bulk, vacancies, and dislocations. The different lifetimes for screw and edge dislocations in Fe single crystals were reported in detail in [48,49]. According to this work, screw dislocations exhibit larger specific trapping rates ν_D_ and lifetimes than edge ones. Similarly, to vacancies in the previous section, it can be shown that the minimum dislocation density detectable by PAS lifetime spectroscopy is ρ_D_ ~10^12^ m^−2^. On the other hand, if ρ_D_ ~10^16^ m^−2^, almost all positrons are trapped at dislocations (saturated trapping), and only the contribution of the trapped positrons is resolved in the PAS spectrum.

Positrons may be trapped in metals also by a grain boundary (GB). Nevertheless, trapping in GBs is likely only when the mean linear dimension of grains does not exceed a few μm. This means the grain size is comparable to (or smaller than) the positron diffusion length L+, and some fraction of positrons have a chance to reach a GB by diffusion motion. The transport of a positron to GBs limits substantially the positron trapping rate in GBs [51].

The relative values of the energy value for the ground state of a delocalized positron in different materials are different. This makes for possible positron trapping in a precipitate with a lower level of positron ground state energy. An excellent review paper about positron applications in radiation-induced defect studies has been published [52]. During the last few decades, many additional destructive and non-destructive methods have been applied with the aim of making progress in the characterization of complex information about defects’ formation and their annealing and/or reannealing [53]. It is necessary to mention that some effects (neutron embrittlement and annealing of defects) are occurring at the same time, and the final microstructure depends on the level of neutron flux/fluence and the temperature of the treatment. Based on previous studies combining Mössbauer spectroscopy (MS) [54,55], positron annihilation spectroscopies (PAS) [56,57,58,59,60,61,62,63,64,65,66,67,68,69,70], and transmission electron microscopy (TEM) [62,64,68], we recognized that the PALS, as well as CDBS measurements, can yield substantial information about types and concentration of vacancies and their behavior after annealing [63,64,65,66,67,68,69,70,71,72,73,74,75]. Positrons trapped in open volume defects, i.e., radiation-induced vacancies, dislocations, microvoids, etc., annihilate with a lower probability than in the perfect area of the same material; thus, the positron lifetime grows with the size of the defects. The production of vacancy defects is directly attributed to radiation damage [59,63]. This was confirmed also by precise TEM studies [76,77]. Nevertheless, the interpretation of results depends on several uncertainties, and the reproducibility could be limited. Generally, we assume that specimens are homogenous. This condition, which is not fully fulfilled in the case of commercial RPV steels, is important for the application of the standard trapping model (STM) [78] in the evaluation of PALS measurements. In inhomogeneous specimens, positrons can be partially attracted by other trapping centers and the various implantation sites, which affects the positron data [79,80]. Therefore, the diffusion-trapping model (DTM) [81] was developed and successfully applied [82,83]. The DTM was several times improved, and its application for the pulsed positron beam technique is described in detail in [84]. The pulsed low-energy positron system (PLEPS) [85,86,87], which enables the measurement of irradiated specimens if their activities are lower than 1 MBq, was applied also for radiation degradation of RPV steels, although the pick/background ratio became significantly worse. This technique enables the depth profiling study of specimens (in the case of steels up to 550 nm) and can reduce the ^60^Co radiation contribution to the lifetime spectra to a minimum.

## 3. Investigated Specimens, Experimental Treatment, and Experimental Techniques

In this work, Russian RPV steel specimens from 15Kh2MFAA (from Greifswald Unit 7, producers: Atomstroyexport JSC, Moscow, Russia & Škoda Works, Plzen, Czech Republic) commercially used in VVER-440 V-213 type reactors including the Mochovce 34 NPP, Slovenské elektrárne in Slovakia (today under physical start-up) are studied from the perspective of radiation damage and structural recovery after the annealing [88]. The investigated material was cut into 10 pieces, creating five pairs of specimens with dimensions of 10 mm × 10 mm × 0.2 mm. The specimens were polished in a mirror-like way to create perfect surface conditions before the surface radiation load. The chemical composition of the investigated specimens of 15Kh2MFAA steel was shown above in Table 1.

The radiation damage was experimentally simulated by proton irradiation (H^+^) in a 500 kV implanter of the Research Centre for Ion Beams and Plasma Technologies SlovakION within the Slovak University of Technology in University Science Park CAMBO Trnava (Slovakia) [89]. Each specimen was implanted (irradiated) with protons with an energy of 500 keV and a hydrogen fluence of 1 × 10^18^ cm^−2^. The total radiation damage achieved a value of up to 1 DPA. The layers were implanted at up to 3 µm according to the SRIM calculation for Fe-2.5Cr-0.6Mo implantation by H ions, and the highest level of radiation damage was at a depth of ~2.3 μm (Figure 2).

The investigated specimens were observed by positron annihilation techniques due to their strong sensitivity to small vacancy defects formed during the process of irradiation/implantation. The radiation load affected mostly surface and subsurface layers, and therefore slow positrons were firstly applied for verification of a defect accumulation and observation of a defect depth profile up of to 520 nm. For that, a pulsed low-energy positron system (PLEPS) with the high-intensity positron source NEPOMUC at the FRM-II reactor in Technical University of Munich, Garching, Germany [90,91] was used.

Later, positron annihilation lifetime spectroscopy (PALS) and coincidence Doppler broadening spectroscopy (CDBS) [92] with a conventional positron source of ^22^Na (Institute of Nuclear and Physical Engineering, Slovak University of Technology, Bratislava, Slovakia) were applied at the Institute of Nuclear and Physical Engineering, Bratislava. The measurements were performed with expectations that only ~20% of positrons can provide a measurable signal from the implanted region up to 3 μm [93], while these techniques investigate materials up to 150 µm.

A PALS spectrum with a total count of ≈10^6^ was measured by three Hamamatsu H3378 photomultipliers tubes coupled with BaF2 scintillators (Hamamatsu Photonics, Hamamatsu City, Shizuoka Pref., Japan) and powered by Ortec 556 HV sources (ORTEC/AMETEK, Oak Ridge, Tennessee, USA) in an air-conditioned box (Figure 3a) at room temperature. The START signal for the lifetime measurement is ~1.2 MeV gamma photons emitted by the positron source ^22^Na. The STOP signal, the annihilation energy of 511 keV, terminates the timing of the positron lifetime measurement, which is directly recorded and stored by DRS4 digitizer chip (RADEC, Koblenz, Switzerland). The own software QtPALS (Petriska, M., Institute of Nuclear and Physical Engineering, Slovak University of Technology, Bratislava, Slovakia) was used for pulse processing and spectrum construction. The data were treated by LT 9 software (Kansy, J., Institute of Physics and Chemistry of Metals, Katowice, Poland).

CDB techniques [94] observed spectra with about 10^6^ counts by two HPGe detectors GC2019 (ORTEC/AMETEK, Oak Ridge, Tennessee, USA) with Gaussian resolution function 1.9 keV at 1.33MeV (Figure 3b) cooled by the cooling system Cryo-JT. As a high voltage source, a dual high-voltage power supply Canbera3125 (Canberra Industries, Inc./Mirion Technologies, Inc., Meriden, CT, USA) is used. The standard multi-channel analyzer and spectra amplifier was replaced by digital components—DAQ Adlink-PCI (ADLINK Technology, Inc., Taoyuan City, Taiwan). The momentum window for a calculation of the S parameter is |pL| < 2.5 × 10^−3^m_0_c, and for the W parameter, it is 15 × 10^−3^ m_0_c < |pL| < 25 × 10^−3^ m_0_c. Pulse amplitudes were collected and evaluated by own software – QtPALS and QtCDB (Petriska, M., Institute of Nuclear and Physical Engineering, Slovak University of Technology, Bratislava, Slovakia).

After the observation of the radiation damage, the specimens were gradually annealed at 200, 300, 400, 450, 475, 500, and 525 °C for detection of the optimal annealing temperature in the process of structural recovery. The post-irradiation annealing was performed in a compact vacuum tube furnace MTI GSL-1800x (MTI Corporation, Richmond, CA, USA). The specimens were annealed for 2 h in a ~6.4 × 10^−9^ vacuum bar and after that cooled down in the air for about 45 min. The specimens were then observed by the PALS technique, whereas the annealing process affected the whole depth of the specimen.

## 4. Results

### 4.1. PLEPS Measurement

PLEPS measurement of one investigated specimen was used to confirm the ability of protons to deliver significant lattice damage and observation of the defect depth profile up to 520 nm. The specimen in the as-received state and after the implantation were compared in terms of defect size and defect presence in the surface and subsurface layers.

Figure 4 shows that positron lifetimes in bulk (LT_1_) of the as-received specimen remained approximately at the same level through all measured depths, which is in perfect agreement with expectations because positron annihilation in bulk should remain the same independently of the state of the material. However, LT_1_ for proton-irradiated specimens compared to the non-irradiated ones increased from 140 ps to 160–170 ps. A slight increase of the positron lifetime LT_1_ in our measurements compared to positron lifetime LT_1_ in pure Fe (~107 ps [95]) might be attributed to the fact that these spectra were also decomposed into two components instead of the usual three components; thus, the DTM model was applied.

Because of the two components’ decomposition, we believe the evaluation software attributed some annihilations to minor lattice defects such as dislocation and monovacancies to the annihilation in bulk, which resulted in the increase of LT_1_. The same explanation applies also to the ~20–30 ps difference between the non-irradiated and the irradiated states—the presence of larger defects in lattices after proton irradiation led to their attribution to annihilation in bulk and therefore the increase of LT_1_ compared to the as-received state.

The intensity of annihilation in bulk (I_1_) for the depth range of 150–520 nm for the as-received state remains constant at values around 90%; only for depths of 50 and 90 nm is I_1_ is somewhat lower. This decrement is naturally accompanied by a proportional increase in the intensity of annihilation in defects (I_2_), as seen in Figure 5. This can be attributed to the fact that positrons after implantation and thermalization move randomly, and during this process, they can also return to the entrance surface.

“Positrons can be trapped at the surface where the electron density is lower than in the bulk. Additionally, they can be trapped at surface defects or positronium, i.e., a bound state between positron and electron can be created” [96]. Another factor that might be contributing is residue defects from specimen preparation—ones that remained even after polishing. All of this causes the I_1_ value to be lower and the I_2_ value to increase at depths closer to the surface than in the bulk (Figure 5). This effect is even more significant for proton-irradiated specimens. As seen in Figure 4, I_1_ decreases from ~95% for depths deep in the bulk to 45% for depths close to the surface (also accompanied by a proportional increase in I_2_ in Figure 5). We believe that this significant decrease in I_1_/increase in I_2_ for proton-irradiated specimens was caused mainly by storage of the specimens (creation of an oxidation layer on the surface, see below) and by manipulation and transportation of specimens during proton irradiation and PLEPS measurements (surface damage).

As seen from Figure 5, positron lifetimes in defects (LT_2_) for depths of 50, 100, and 150 nm is 20–40 ps higher for the proton-irradiated specimens compared to the as-received state. At depths this close to the surface, the possibility that this disproportion would be created by proton irradiation is excluded because, as shown by SRIM simulations, the lattice damage starts at depths of ~100–150 nm (Figure 5). This increase in LT_2_ was probably caused by the creation of an oxidation layer on the surface of the specimen. The assumption of the oxidation layer creation is supported by the results from other papers focused on the study of corrosion-related defects by slow positron beam. These papers reported similar values of LT_2_ in the surface oxidation layer [97,98]. However, the creation of the surface oxidation layer was expected because the period between specimen preparation and PLEPS measurements of proton-irradiated specimens on this occasion was prolonged because of various problems with the FRM-II reactor to more than two years. Figure 5 also shows that the surface oxidation layer reaches a depth of around 200 nm, where the positron lifetime in defects (LT_2_) for proton-irradiated specimens exhibits the same values as for the non-irradiated state (without the oxidation layer).

For depths starting at ~270 nm, a progressive increase in LT_2_ for the proton-irradiated state is clear compared to the non-irradiated state. This progressive increase in LT_2_ is in perfect agreement with SRIM simulations because, as seen in Figure 2, the number of vacancies produced starts to increase at this depth. At maximum depth, LT_2,_ reaches ~460 ps, which indicates the presence of vacancy clusters containing more than 40 vacancies [99]. It is important to note that the maximum depth of positrons reached by PLEPS is, as seen in Figure 4 and Figure 5, around 0.52 μm, but according to SRIM simulations, the maximum lattice damage is at 2.3 μm; therefore, even bigger vacancy clusters are expected to form deeper in the material. Unfortunately, we were unable to probe deeper into the specimen because 0.52 μm is the maximum depth that can be reached by positrons in PLEPS. By using 500 keV protons, we hit the spot where this energy is too low for effective usage of conventional PALS and simultaneously too high for PLEPS to see the overall lattice damage. Nevertheless, PLEPS data confirmed that lattice damage was introduced to the material by proton irradiation.

### 4.2. CDB Measurement

The CDB measurements using standard positron energy spectra were preliminarily applied only for five as-received and five implanted specimens as confirmation of PLEPS results and the test of sensitivity to the observed defects by a conventional positron source. The CDB results characterizing the material at a depth of 150 µm are shown in the S-W diagram (Figure 6), where the W parameter describes positron annihilation with core electrons (annihilation in bulk) and the S parameter with valence electrons located mostly in open volume (annihilation in vacancy defects). These two parameters are extracted from each spectrum, and each carries different information about the measured materials [4].

As seen in Figure 6, proton-irradiated specimens exhibit a significant increase in the S parameter and a decrease in the W parameter compared to the non-irradiated specimens. The total amount of annihilating positrons is fixed, and in general, an S parameter increase is necessarily accompanied by a W parameter decrease. The S parameter can be affected by both the number density and the size of vacancy-type defects [100], so in our case, this enhancement of the S parameter indicates defect accumulation in the structure due to the implantation. These results are in good agreement with the PLEPS results above, and they also indicate that conventional positron sources can be useful for the detection of implantation-accumulated defects in our structure.

It is important to note that the data’s variance for all as-received specimens is 0.35%, and for all irradiated ones, it is 0.11%, i.e., the scattering of the measured data is very low. This proves that the results are not very sensitive to material inhomogeneity, clusters of carbides, disparities in specimen preparation, or inaccuracies during experimental measurements and during proton irradiation of the specimens. Therefore, the standard trapping model [78] can be applied for the evaluation of the PALS spectra.

The average S parameter of the as-received specimens is 0.6108 ± 0.0007, and the average W parameter is 0.24998 ± 0.0007. The average CDB parameters changed due to the implantation, and they are S = 0.617 ± 0.0003 and W = 0.2444 ± 0.0003. The implantation caused the S parameter to increase about 1.01 times, which indicates an accumulation of very small defects (mono- or di-vacancies).

### 4.3. PALS Measurement

All five investigated specimens were observed by the PALS technique in the as-received state, after the implantation, and after the gradual annealing. They showed good homogeneity in structure, and the response to the experimental loads was also similar, as can be seen in Figure 7. Specimen no. 5 is measured only up to the annealing temperature of 500 °C due to the specimen sustaining mechanical damage during the experiment.

PALS data were evaluated in the program LT9 by the standard trapping model [90] and separated into two or three components including lifetimes (LT) and intensities (I). For the first lifetime (LT_1_), the bulk was fixed to 100 ps (calculated values for a pure iron range from 97 ps [95] to 110 ps [48]) for better comparison of the second components for all measurements. The second component involves small defects and can describe their size as proportional to a lifetime (LT_2_, mostly dislocations and vacancies) and their concentration which is proportional to the intensity (I_2_). The third component (LT_3_, I_3_) was found only for implanted specimens and annealed specimens at temperatures out of range (300–500) °C, where we expected the most visible change in structure. The LT_3_ was on average 1.4 ± 0.5 nm with I_3_ of approx. 1% and describes annihilation in the source or infight annihilation not fully compensated during the process of fitting.

The LT_2_, I_2,_ and positron Mean Lifetime values (MLT, describing the whole spectra while minimizing the influence of the data treatment) are presented in Figure 7 for all measurements. For the individual treatments of specimens, the average of LT_2_ and I_2_ as well as their standard deviations were calculated considering the uniformity of the specimens and their good structural homogeneity. These average values are described below in more detail.

In the as-received state, the average LT_2_ = 158 ps ± 3 ps probably represents dislocations together with the smaller presence of mono-vacancies. The average intensity of as-received specimens was 59.7% ± 2.5%. The Mean Lifetime for the as-received specimens was between 134 and 137 ps, and the average value was 135 ps ± 2 ps. It is the lowest value for all our measurements, which indicates the lowest total defect volume (Figure 8), as was expected.

After the implantation, LT_2_ increased, with the average value of 188 ± 3 ps representing a mixture of mono-vacancies and di-vacancies with intensity (I_2_) 42.8 ± 1.6%. The third lifetime appeared with average values of 1 544 ± 820 ps and intensity of 0.4 ± 0.1%. The MLT of the implanted specimen was between 142 and 146 ps, and the average value was 143 ± 2 ps. This is a much higher MLT value than for the as-received specimens. The MLT change was 8 ± 4 ps, which proves there was defect accumulation due to H^+^ implantation in the specimen. The mono-vacancies partially changed into di-vacancies during the process of irradiation. Although previously found dislocations did not disappear from the structure, only the positrons were more attracted and trapped in the larger defects, di-vacancies, which could be near the dislocations [45].

The second components of PALS spectra and MLT were progressively changed after the gradual experimental annealing at 200, 300, 400, 450, 475, 500, 525, and 550 °C, as is seen in Figure 7. The annealing temperature of 200 °C did not show significant change from the perspective of MLT, 142 ± 1 ps, but there were found mostly mono-vacancies, probably together with dislocations again. The average value of the second component for the specimens annealed at 200 °C is LT_2_ = 165 ± 3 ps and I_2_ = 59.1 ± 1.5%. The existence of the third component affected the MLT value, and thus its change could probably not be so evident. The average values for third component are LT_3_ = 992 ± 190 ps and I_3_ = 0.4 ± 0.09%.

The specimens annealed at temperatures from 300 to 500 °C showed a small decrease in defect presence compared to the implanted data as well as the specimens annealed at 200 °C. The average MLT values for all annealed specimens are similar, being 140 ± 2 ps (Figure 7). Only the specimens annealed at temperatures between 475 and 500 °C had MLT values of 139 ± 1 ps, which is a practically negligible difference. However, the temperature of 475 °C is commercially used for the recovery of reactor steels in older VVER reactors. Our radiation-induced changes had much lower levels and were only present in the thin layer of the specimens, and thus the change here is not so significant.

The second components for specimens annealed at temperatures between 300 and 500 °C are almost the same and showed the presence of mostly mono-vacancies together with dislocations: LT_2_ = 169 ± 2 ps and I_2_ = 57.3 ± 0.9% (at 300 °C), LT_2_ = 169 ± 2 ps and I_2_ = 56.8 ± 1.4% (at 400 °C), LT_2_ = 168 ± 2 and I_2_ = 57.6 ± 1.3% (at 450 °C), LT_2_ = 167 ± 2 ps and I_2_ = 57.1 ± 1.8% (at 475 °C), LT_2_ = 166 ± 2 ps and I_2_ = 56.68 ± 0.9% (at 500 °C).

The MLT for specimens annealed at higher temperatures than 500 °C indicates an increase in the defect presence at a more significant rate than during the implantation. The second component for specimens annealed at 525 °C has average values LT_2_ = 176 ± 3 ps (mono-vacancies) and I_2_ = 57.3 ± 0.9%. The second component for specimens annealed at 550 °C is LT_2_ = 177 ± 2 ps (mono-vacancies) and I_2_ = 56.3 ± 0.6%. These specimens have also third components with average values: LT_3_ = 1 228 ±150 ps and I_3_ = 1.19 ± 0.2 (at 525 °C), LT_3_ = 1 865 ± 180 ps and I_3_ = 1.3 ± 0.2 (at 550 °C).

From the positron data, the defect concentrations and defect volume were calculated according to equations published in [101,102,103] and presented in Figure 8. The average values of defect concentration and defect volume for the individual treatments are seen in Figure 8. The calculation was applied only for the material visible by positrons up to a depth of approximately 150 µm, which is a volume of around 0.012 cm^3^ with ≈ 10^21^ atoms.

The average defect concentration (C_D_) decreased about ΔC_D_ = − (2.85 ± 0.89) ppm after the implantation due to the formation of larger defects (from dislocation with mono-vacancy to mostly di-vacancy). However, the average defect volume (V_D_) observed by positrons increased by ΔV_D_ = + (2.88 ± 1.05) × 10^−8^ cm^3^ due to the implantation.

After annealing at 200 °C, the size of defects decreased, and the defect concentration increased back by ΔC_D_ = + (3.84 ± 0.83) ppm compared to the implanted specimens and by ΔC_D_ = + (0.99 ± 0.73) ppm for to the as-received specimens. This indicates the presence of more defects than in the two previous stages of the specimens. The average defect volume decreased by ΔV_D_ = − (2.00 ± 0.97) × 10^−8^ cm^3^ compared to the implanted specimens due to partial recombination of di-vacancies and a shift of the average defect size from di-vacancy to a vacancy. The change of the defect volume compared to the as-received specimens is ΔV_D_ = + (0.88 ± 0.72) × 10^−8^ cm^3^, and thus the annealing did not recover the material.

The decline of defect volume due to annealing was the biggest at 200 °C, although the defect volume still gradually decreases up to 500 °C, almost only within the error bars. However, the absolute lowest values for defect concentration and defect volume within the annealed specimens seem to occur at temperatures 450 and 475 °C. The values ΔC_D_ ≈ − 3.19 ppm and ΔV_D_ ≈ − 2.57 × 10^−8^ cm^3^ showed the biggest decrease compared to the implanted specimens for both temperatures (the same values). The changes in the values for 450 and 475 °C compared to the as-received specimens are ΔC_D_ ≈ + 0.34 ppm and ΔV_D_ ≈ + 0.30 × 10^−8^ cm^3^, which shows that no full recovery of the structure occurred due to the annealing.

At the higher temperatures of 500, 525, and 550 °C, the defect concentration and the defect volume visibly increased more than ΔC_D_ ≈ 0.85 ppm and ΔV_D_ ≈ 0.75 × 10^−8^ cm^3^ compared to the as-received specimens, but the defect volume is still lower than for the implanted specimens due to smaller defects. This could be due to thermal strain and the formation of new vacancies due to structural changes (precipitates, defect mobility, thermo-vacancies).

## 5. Discussion

Degradation of design materials of nuclear installations via aging or neutron embrittlement is a permanent and long-term studied nuclear-relevant problem. In addition to conventional methods (destructive tests), several non-destructive techniques including positron annihilation were involved in these studies. The reproducibility of PAS results after more than 30 years of application of these techniques worldwide (with the sometimes-disputable origin of specimens or only having a few specimens or based on laboratory-prepared binary alloys) was discussed and analyzed in this paper.

Due to difficulties related to the traditional neutron irradiation experiments (expense, time consumption, special equipment requirements, and radioactivity), the first research aimed to study proton irradiation of 15Kh2MFAA steel as a potential surrogate of neutron irradiation in experiments related to LTO. Proton irradiation provides a faster, cheaper, less dangerous, and repeatable way to investigate basic radiation damage processes taking place in RPVs. 15Kh2MFAA is the reactor pressure vessel steel that was used as the structural material for the Russian VVER-440/213 reactors. Specimens of 15Kh2MFAA steel obtained from the surveillance program of the Mochovce 34 NPP were irradiated by light ions (500 keV protons) to emulate radiation damage from neutrons. After irradiation, specimens were annealed at various temperatures to determine the level of lattice recovery at each temperature. Three different non-destructive positron annihilation-based techniques (positron annihilation lifetime spectroscopy—PALS, coincidence Doppler broadening spectroscopy – CDBS, and pulsed low-energy positron system—PLEPS) were used to investigate and characterize the microstructural properties of the 15Kh2MFAA steel. The characterization was based on the determination and comparison of the size and concentration of vacancy-type defects in the material.

In this study, 500 keV proton implantation delivered only slight lattice damage to the structure, and therefore the annealing experiment did not provide satisfactory results. However, we were able to detect slight differences in positron data for the as-received state, irradiated state, and annealed state at all investigated temperatures.

PLEPS and CDB results provided evidence for changes in microstructure due to the implantation. The PALS data indicate that the implantation process can be also observed by conventional methods, although the data changes are often very indistinct and should be confirmed or completed by another technique, such as PLEPS and CDB in this study. The increase of MLT due to the implantation was ≈8 ps, which represents a small difference, but enough for an indication of a change in structure and defect presence.

The annealing experiment observed by the PALS technique found smaller changes, but they progressed with the increase of the annealing temperature. While the temperature of 200 °C showed the biggest decrease in the defect volume after the implantation, the absolute lowest values of defect volume and concentration were found at the temperatures of 450 and 475 °C. In terms of the Mean Lifetime, annealing between 475 and 500 °C seems to be the best for structural recovery.

It is known from accumulated research that carbon atoms within the matrix play a key role in terms of vacancy cluster formation in Fe-C alloys and steels. Carbon and vacancies strongly interact and form stable carbon-vacancy (CV) complexes such as CV, C_2_V, C_4_V_2_, and bigger ones. These CV complexes act as traps for radiation-induced vacancies, leading to the enhanced formation of vacancy clusters [104,105,106,107]. Therefore, the small recovery observed in the temperature interval of 200–300 °C could be explained by the decomposition of the small carbon-vacancy pairs or complexes. These findings correspond well with the available literature, where various authors note that the dissolution of vacancy complexes occurs at this temperature range in steels and Fe-C alloys [108,109,110,111]. However, this phenomenon of annealing CV complexes probably does not play a key role in terms of lattice recovery of VVER RPVs after irradiation because of the normal operating temperature (270–300 °C), and therefore it is expected that those types of defects will recombine shortly after their origin.

Furthermore, the PALS data for the annealing at 500, 525, and 550 °C proved that higher temperatures affect the structure via thermal structural changes. The increase in positron lifetime above 475 °C corresponds to the nucleation and growth of metal carbide/nitride precipitates, with a majority of the precipitates being MC, M_7_C_3_, and M_23_C_6_ carbides [96,112,113,114]. The carbide precipitation begins at 450 °C with vanadium-rich carbides (MC). Simultaneously, or at somewhat higher temperatures iron–chromium carbides (M_3_C) precipitate. These are replaced by M_7_C_3_ carbides at higher temperatures [113]. With the increase in the number density and the size of precipitates, the associated precipitate–matrix interface area increases, and hence, the interfacial open volume defects increase. The increase in the lifetime could be also due to vacancy defects inside the precipitates [112]. The PALS results for higher temperatures confirmed that the ideal annealing temperature for 15Kh2MFAA is up to 475 °C. Similar behavior of positron lifetime in this region for RPV or model steels has been reported by various authors [112,115,116,117].

We assume that open volume defects as vacancies and interstitial points are mobile and recombine together at the normal operating temperatures of most RPVs. However, it is also expected that they can interact with solute atoms. The key interstitial impurity in Russian RPV steel is carbon [118,119]. The partial or complete trapping of self-interstitial points by C solutes will cause heterogeneous cluster nucleation and fine cluster distribution. Furthermore, in steels containing residual levels of elements such as copper, which are in super-saturated solution, radiation-enhanced diffusion will occur at these temperatures, which leads to the formation of small clusters, which can again harden the matrix. Generally, thermal treatment together with neutron irradiation led to a microstructure consisting of small clusters (<5 nm in diameter) which create obstacles to the free movement of dislocations, thereby producing an increase in the yield stress, hardness, and ductile-brittle transition temperature of the material.

It was shown that PLEPS can see the formation of solute clusters during irradiation of RPV steels, resulting in the depletion of Cu and P in the matrix in the first period of irradiation [86,95,118]. Some alloying elements (for example, Ni) can slightly retard this depletion. After this first period, both techniques registered no significant changes connected to increased neutron treatment. On the contrary, the positron lifetimes decreased, probably due to long-term thermal treatment at the level of 280 °C. Using PALS, the most effective region (450–475 °C) of thermal treatment was clearly shown.

Positron annihilation techniques can be applied for the development of new types of steel for advanced nuclear facilities such as fusion reactors or spallation neutron sources [120]. They are particularly useful for the evaluation of the effectiveness of post-irradiation thermal treatments [78]. The future is open to new investigative methods, and an application of a scanning positron microscope for testing RPV steel would surely be a good method [121].

## 6. Conclusions

The present paper reports a multi-specimen positron annihilation study of reactor pressure vessel steel 15Kh2MFAA in the as-received state, after proton implantation, as well as after isochronal annealing at temperatures (200–550) °C. Based on the results, the overall reproducibility of the obtained data was determined to be better than 2 ps in terms of the positron Mean Lifetime. The proton implantation, performed as an experimental simulation of neutron exposure, led to apparent differences in positron lifetime and Doppler broadening (of the annihilation gamma line) spectra. The presence of radiation-induced defects was observed via techniques utilizing slow positron beams as well as via techniques based on radioisotope positron sources.

According to our experimental results, it is possible to conclude that all used positron annihilation techniques registered the accumulation of small open-volume defects (mostly mono- and di-vacancies) due to the simulated irradiation (proton implantation), with the increase of the defect volume being ΔV_D_ = 2.88 × 10^−8^ cm^−3^.

The annealing experiments were performed on radiation-loaded specimens in the temperature range from 200 to 550 °C and analyzed. The best defect recovery was found at temperatures between 450 and 475 °C, but the defect concentration was still higher there, about ΔC_D_ = 0.34 ppm higher than in the as-received specimens.

## Figures and Tables

**Figure 1 materials-15-07091-f001:**
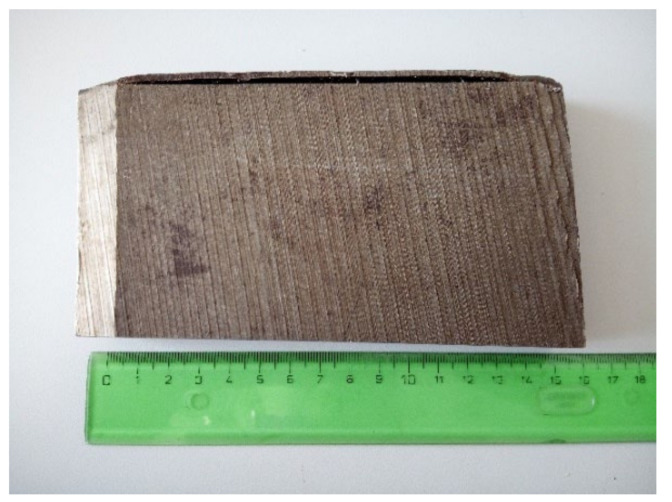
Illustrative cross-section from VVER-440 RPV (Original not-irradiated peace from VVER-440 V-213 RPV—Greifswald Unit 7).

**Figure 2 materials-15-07091-f002:**
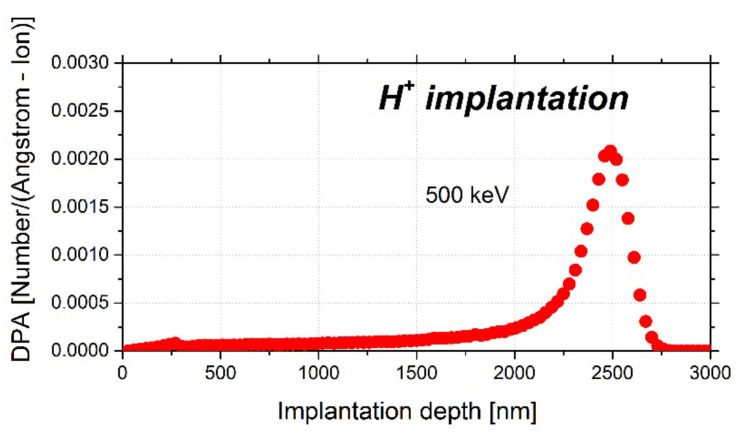
SRIM calculations of DPA for 500 keV H^+^ ions incident on Fe-2.5Cr-0.6Mo. The detailed K-P calculation was used with a total number of ions in the calculation of 5 × 10^4^.

**Figure 3 materials-15-07091-f003:**
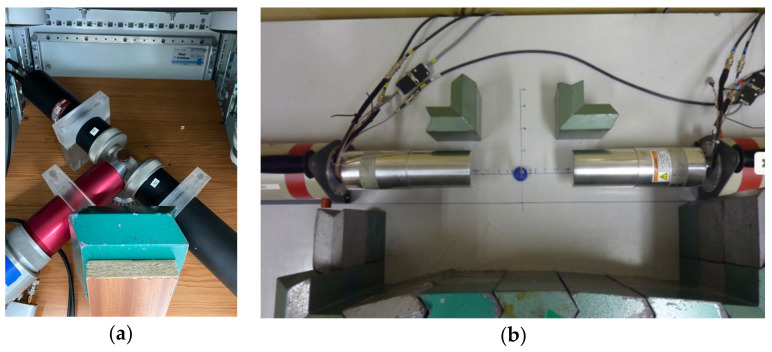
Air-conditioned unit setup for PALS (**a**) and CDB instrument in Institute of Nuclear and Physical Engineering, Bratislava (**b**).

**Figure 4 materials-15-07091-f004:**
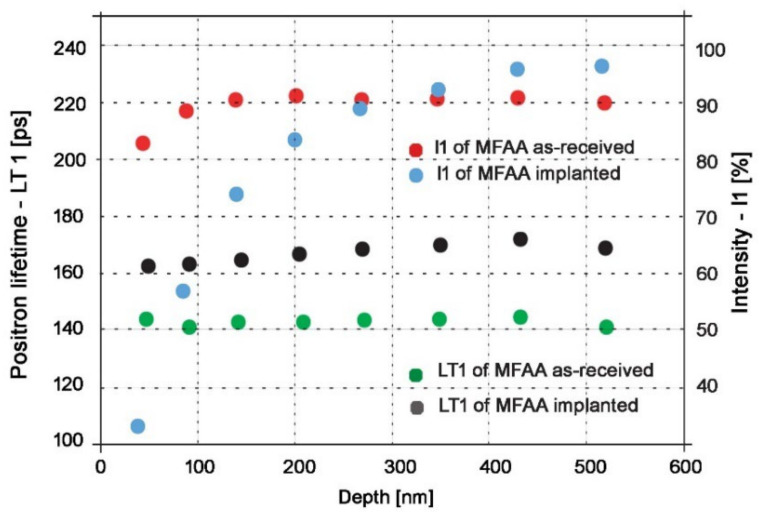
Positron lifetimes (LT_1_) and intensity of annihilation (I_1_) in bulk for various energies of incident positrons (penetration depth) for both non-irradiated and proton-irradiated specimens.

**Figure 5 materials-15-07091-f005:**
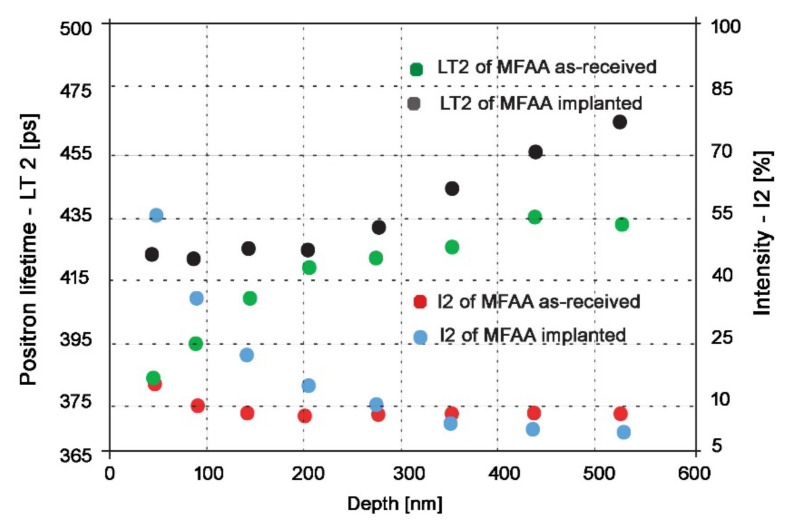
Positron lifetimes (LT_2_) and intensity of annihilation (I_2_) in defects for various energies of incident positrons (penetration depth) for both non-irradiated and proton-irradiated specimens.

**Figure 6 materials-15-07091-f006:**
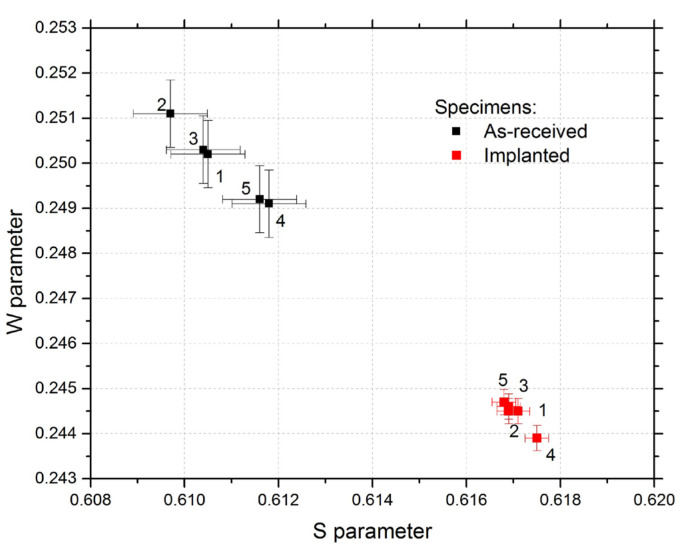
W parameter as a function of the S parameter for the as-received and proton-irradiated state.

**Figure 7 materials-15-07091-f007:**
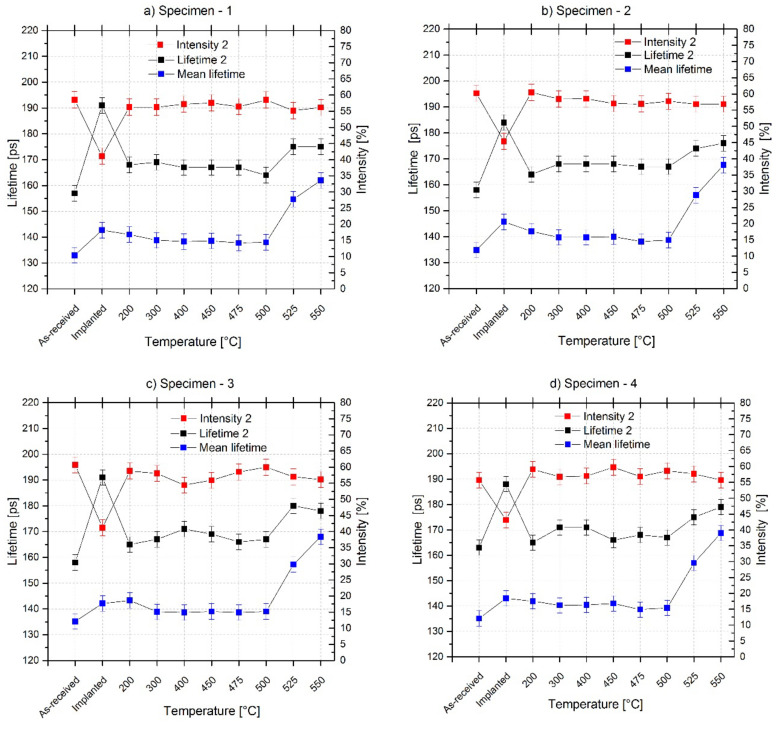
PALS data for as-received, implanted, and annealed specimens: Lifetime 2 proportional to the size of the defects; Intensity 2 proportional to defect concentration; and Mean lifetime describing the defect presence in the specimens.

**Figure 8 materials-15-07091-f008:**
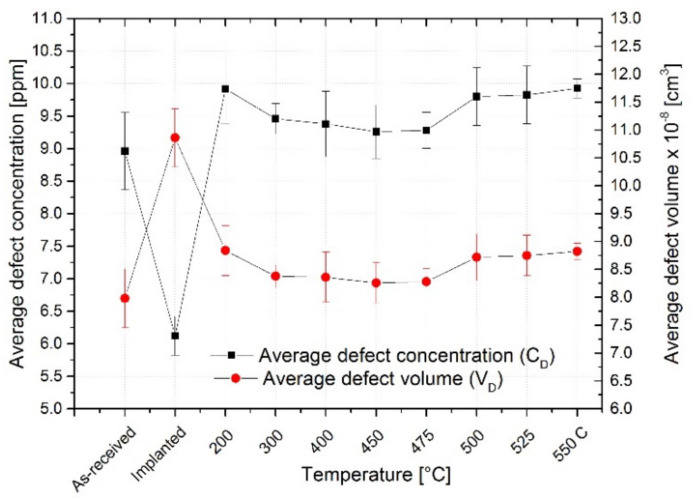
Average values of defect concentration and defect volume for the as-received, implanted, and annealed specimens at 200, 300, 400, 450, 475, 500, 525, and 550 °C.

**Table 1 materials-15-07091-t001:** Chemical composition (in wt.%) of 15Kh2MFA and 15Kh2MFAA RPV steels.

Steel	C	Mn	Si	P	S	Cr	Ni	Mo	V	Cu
15Kh2MFA	0.13	0.30	-	max	max	2.50	max	0.60	0.25	max
	0.18	0.60	-	0.020	0.025	3.00	0.40	0.80	0.35	0.30
15Kh2MFAA	0.11	0.30	0.17	max	max	2.00	max	0.60	0.25	max
Investigated steel	0.16	0.60	0.37	0.012	0.015	2.50	0.40	0.80	0.35	0.10
	Additions:	**As**	**Sb**	**Sn**	**P+Sb+Sn**	**Co**	**P**	**S**	**Cu**	
		0.10	0.005	0.005	0.015	0.02	0.012	0.15	0.08	

**Table 2 materials-15-07091-t002:** Calculated lifetimes τ and binding energies EB of positrons trapped in the core region of dislocation line and defects associated with dislocation [43,47,48].

Annihilation Site	τ	EB
	(ps)	(eV)
Fe: defect-free structure	97–110	
mono-vacancy	179	−2.39
di-vacancy	195	
vacancy with hydrogen	146	
Fe: edge dislocation line	117	−0.28
jog on the edge dislocation line	117	−0.11
vacancy on the edge dislocation line	140	−0.92
di-vacancy on the edge dislocation line	117	−0.07
Fe: screw dislocation	114	−0.22
vacancy in screw dislocation	174	−2.35

## Data Availability

The data presented in this study are available on request from the corresponding author.

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
