# Peer review of "Positron Annihilation Study of RPV Steels Radiation Loaded by Hydrogen Ion Implantation"

_materials, 2022, doi:10.3390/ma15207091_

Round 1

Reviewer 1 Report

This paper shows positron annihilation study of RPV steels radiation embrittlement simulated by hydrogen implantation. Authors presented in Introduction that generally RPV irradiation embrittlement are caused by matrix microstructure damage, Cu-rich precipitates, MnNi-rich precipitates, and P segregation etc. However, positron annihilation method does not show which kind of precipitates forms and whether P segregation occurs or not. I think that it is necessary to investigate microstructural change after irradiation using electron microscopies. Therefore, I am not sure this method is suitable for study on degradation of nuclear materials after irradiation.

Author Response

English has already been improved by using a program Grammarly.

Information about Cu-rich and Ni/Mn/Si-rich precipitates were mentioned in the text from cited papers and cannot be directly seen by PAS. However, the damage due to ion implantation is effectively studied using PAS technique for more than 50 years [65,66,71,74,10,115,120,121].

Recent research - reference papers [14,15], newly added during the process of revision, show that the vacancy-defects significantly affect formation of precipitates.

The changes of mechanical properties as embrittlement due to irradiation as well as electron microscopy techniques has already been well and wide published [26-28, 67,68]. However, we tried to clarify the irradiation-induced change from the microstructural defect side, where positron techniques provide unique view for that.

The relation between defects and precipitates can be effectively measured by Doppler broadening of positron spectra partially described in the text, which can investigate the chemical surrounding of the defects. The detailed DBS analyses of these materials is still ongoing in several laboratories.

Reviewer 2 Report

The papers deals with the evaluation of neutron irradiation induced damage of ferritic steels. The technique used is positron annihilation that is well recognized as an effective technique to this purpose. The paper is well structured and the results are well described as well as the discussion

Author Response

Thank you for your review.

Reviewer 3 Report

The manuscript entitled "Positron annihilation  study of RPV steels radiation embrittlement simulated by hydrogen ions implantation" is appropriate for publication in the Journal of Materials after some corrections, for example: 

1. The introduction section should be summarized and focused on the main purpose of the research work.

2. The results of this paper should be confirmed by other methods such as electron microscopy.

Author Response

1. English has already been improved by using a program Grammarly.

2. "The introduction section should be summarized and focused on the main purpose of the research work."

We accepted your comment.

More information about Cu-rich and Ni/Mn/Si-rich clusters were added in the text and new recently published papers were cited.

3. "The results of this paper should be confirmed by other methods such as electron microscopy."

This remark is basically correct, accurate and fully in line of paper improvement. Unfortunately, we do not have our own data from electron microscopy. However, some electron microscopy research of reactor steels or Fe-Cr alloys has already been applied by another authors. The citation of these publications was added in the text during the process of revision.

Reviewer 4 Report

This paper studies RPV steels under irradiation. There are quite a few points of revision I have to request before the paper is accepted. See my comments below:

1. “„While” Please correct this.

2. Table 1 caption has an error.

3. There are some recent published papers showing the spatial correlation (looks in-pair) between Cu-rich and Ni/Mn/Si-rich clusters in both RPV steels and additive manufactured HT9 alloys under irradiation. I would consider mentioning them and expand the introduction, especially the formation of both clusters is the major contributor to irradiation hardening/embrittlement.

4. Figure 2: please convert the SRIM output into dpa. Also, a lot of the details of the SRIM calculation are missing. What’s the stopping powers for each element in the RPV steel? What’s the model used for the calculation, quick K-P or detailed K-P?

5. Also, I thought the concentration of this work is on the investigation of precipitates and hardening. But PAS is a technique mainly for vacancies. I don’t see how these two are connected within this work. Where are the data or discussion on hardening/embrittlement, which is part of the title of this work??

Author Response

1. "„While” Please correct this."

  •  The sentence with “While” was reformulated.

2. "Table 1 caption has an error."

  • The error was fixed.

3. "There are some recent published papers showing the spatial correlation (looks in-pair) between Cu-rich and Ni/Mn/Si-rich clusters in both RPV steels and additive manufactured HT9 alloys under irradiation. I would consider mentioning them and expand the introduction, especially the formation of both clusters is the major contributor to irradiation hardening/embrittlement."

  • We accepted the reviewer's comment. The mentioned publications were found and cited in the text.

4. "Figure 2: please convert the SRIM output into dpa. Also, a lot of the details of the SRIM calculation are missing. What’s the stopping powers for each element in the RPV steel? What’s the model used for the calculation, quick K-P or detailed K-P?"

  • It was accepted. The figure was replaced and the information about model was added. The SRIM calculation was done only for Fe-2.5Cr-0.6Mo since other elements are under 0.5% in wt.

5. "Also, I thought the concentration of this work is on the investigation of precipitates and hardening. But PAS is a technique mainly for vacancies. I don’t see how these two are connected within this work. Where are the data or discussion on hardening/embrittlement, which is part of the title of this work??"

  • The reviewer is right that the hardening or embrittlement effect was studied here only by indirect way with application of positron annihilation technique. The changes of mechanical properties as embrittlement due to irradiation has already been well and wide published. However, we tried to clarify that from the microstructure side, where positron techniques provide unique view for that.
  • The increase of vacancy presence and accumulation of implanted ions due to implantation were mostly proportional to material embrittlement. It was experimentally proven many years ago and that is described also in the introduction and with citation of reference papers [12,13]. Although, the embrittlement effect is described mostly by precipitates formation, recent published paper [14,15] showed that precipitates are located near the defects and defects affect the size of precipitates, which we also expected in this study.
  • It means that the effect of embrittlement is probably dependent on both – vacancy defects presence and precipitates. Thus, Doppler broadening of annihilation spectra describing the defects together with their chemical surrounding, partly applied also in this paper, is an ideal investigation method should be applied further.

Round 2

Reviewer 1 Report

I understood your data clearly. 

Author Response

Thank you.

Reviewer 4 Report

The revision is almost satisfactory, but I just one more comment on the author response:

1. I still believe this paper is investigating the microstructure, rather than hardening and embrittlement. The title can be very misleading, as no mechanical property data are reported in this work. Of course, microstructural changes affect mechanical properties, but I wouldn’t call it “an indirect way of hardening effects”. Please consider changing the title of this work, as this is solely a microstructural study.

Original comment and response:

Comment: 

"Also, I thought the concentration of this work is on the investigation of precipitates and hardening. But PAS is a technique mainly for vacancies. I don’t see how these two are connected within this work. Where are the data or discussion on hardening/embrittlement, which is part of the title of this work??"

Response: The reviewer is right that the hardening or embrittlement effect was studied here only by indirect way with application of positron annihilation technique. The changes of mechanical properties as embrittlement due to irradiation has already been well and wide published. However, we tried to clarify that from the microstructure side, where positron techniques provide unique view for that.

The increase of vacancy presence and accumulation of implanted ions due to implantation were mostly proportional to material embrittlement. It was experimentally proven many years ago and that is described also in the introduction and with citation of reference papers [12,13]. Although, the embrittlement effect is described mostly by precipitates formation, recent published paper [14,15] showed that precipitates are located near the defects and defects affect the size of precipitates, which we also expected in this study.

It means that the effect of embrittlement is probably dependent on both – vacancy defects presence and precipitates. Thus, Doppler broadening of annihilation spectra describing the defects together with their chemical surrounding, partly applied also in this paper, is an ideal investigation method should be applied further.

Author Response

I agree with the reviewer. The title of the paper has been changed into:

"Positron annihilation study of RPV steels radiation loaded by hydrogen ions implantation ".